# Clinical Features and Disease Progression in Older Individuals with Rett Syndrome

**DOI:** 10.3390/genes15081107

**Published:** 2024-08-22

**Authors:** Jeffrey L. Neul, Timothy A. Benke, Eric D. Marsh, Bernhard Suter, Cary Fu, Robin C. Ryther, Steven A. Skinner, David N. Lieberman, Timothy Feyma, Arthur Beisang, Peter Heydemann, Sarika U. Peters, Amitha Ananth, Alan K. Percy

**Affiliations:** 1Department of Pediatrics, Vanderbilt Kennedy Center, Vanderbilt University Medical Center, Nashville, TN 37232, USA; cary.fu@vumc.org (C.F.); sarika.u.peters@vumc.org (S.U.P.); 2Department of Pediatrics, Neurology and Pharmacology, University of Colorado School of Medicine, Children’s Hospital Colorado, Aurora, CO 80045, USA; tim.benke@cuanschutz.edu; 3Department of Neurology, Children’s Hospital of Philadelphia, University of Pennsylvania, Philadelphia, PA 19104, USA; marshe@chop.edu; 4Department of Pediatrics, Baylor College of Medicine, Houston, TX 77030, USA; 5Department of Neurology, Washington University School of Medicine, St. Louis, MO 63110, USA; rytherrc@wustl.edu; 6Greenwood Genetic Center, Greenwood, SC 29646, USA; sas@ggc.org; 7Department of Neurology, Boston Children’s Hospital, Boston, MA 02115, USA; david.lieberman@childrens.harvard.edu; 8Gillette Children’s Specialty Healthcare, St Paul, MN 55101, USAabeisang@gillettechildrens.com (A.B.); 9Department of Pediatrics, Rush University Medical Center, Chicago, IL 60612, USA; peter_heydemann@rush.edu; 10Department of Pediatrics, University of Alabama at Birmingham, Birmingham, AL 35294, USA; aananth@uabmc.edu (A.A.); apercy@uabmc.edu (A.K.P.)

**Keywords:** Rett syndrome, *MECP2*, old age, clinical severity, disease progression

## Abstract

Although long-term survival in Rett syndrome (RTT) has been observed, limited information on older people with RTT exists. We hypothesized that increased longevity in RTT would be associated with genetic variants in *MECP2* associated with milder severity, and that clinical features would not be static in older individuals. To address these hypotheses, we compared the distribution of *MECP2* variants and clinical severity between younger individuals with Classic RTT (under 30 years old) and older individuals (over 30 years old). Contrary to expectation, enrichment of a severe *MECP2* variant (R106W) was observed in the older cohort. Overall severity was not different between the cohorts, but specific clinical features varied between the cohorts. Overall severity from first to last visit increased in the younger cohort but not in the older cohort. While some specific clinical features in the older cohort were stable from the first to the last visit, others showed improvement or worsening. These data do not support the hypothesis that mild *MECP2* variants or less overall severity leads to increased longevity in RTT but demonstrate that clinical features change with increasing age in adults with RTT. Additional work is needed to understand disease progression in adults with RTT.

## 1. Introduction

Rett syndrome (RTT) [1,2] is a significantly disabling neurodevelopmental disorder primarily, but not exclusively, affecting girls and women, that is caused in the majority of cases (>96%) by pathogenic loss of function genetic variants in the X-linked gene *methyl-CpG-binding protein 2* (*MECP2*) gene [3,4,5]. Over the past twenty years, it has been recognized that long-term survival of people with RTT is both possible and likely [6,7]. Data obtained from a North American database demonstrated that median survival of people with RTT is greater than 50 years of age [8]. This finding stands in marked contrast to the observed survival of the cohort of people with RTT originally identified by Andreas Rett beginning in the 1950s [9]. The observed improvement in survival might relate to factors such as earlier recognition, better management of nutritional concerns, improved physical, occupational, and communication therapies, and better approaches to the associated problems of seizures, gastrointestinal issues, and scoliosis [10,11,12,13,14,15,16,17,18,19,20,21,22].

Although it has been recognized that many individuals with RTT survive into adulthood, limited work has evaluated the clinical features of mature women with this disorder. Lotan et al. [23] reported three women greater than thirty years old in Israel whereas Peron et al. [24] described fifty-six women with RTT ranging in age between 19–49 in Italy. However, the median age of this latter group was 29 years yielding not more than 28 women 30 years of age or greater. Gross motor skills were evaluated in 24 individuals in Denmark, age 30–66 [25], and medical issues and epilepsy in smaller numbers in Norway [26,27].

Because there is a well-established genotype–phenotype relationship in RTT [28], the hypothesis has been proposed that individuals with *MECP2* variants associated with overall milder involvement, (R133C, R294X, R306C, and C-terminal truncations) would demonstrate greater overall survival [28]. The corollary to this is that individuals with more severe variants (R106W, R168X, R255X, R270X and large deletions) would be more likely to succumb earlier such that the percentage of those with mild vs. severe mutations would change with increasing age. Among the milder variants, greater maintenance of ambulation and purposeful hand function and lesser difficulties with seizures and scoliosis had been noted previously across the age spectrum [28]. Although previous work evaluating the association between longevity and *MECP2* variants did not identify such an association [29], the increased survival observed in the US Rett syndrome and RTT-related Disorders Natural History Study (RNHS), comprising sixteen years of longitudinal data from over 1200 people with RTT, prompted a further evaluation of this hypothesis. Additionally, we sought to evaluate whether clinical features were stable or continued to change in older individuals with RTT.

## 2. Materials and Methods

### 2.1. Participants

Participants were enrolled in the Rett syndrome and RTT-related Disorders Natural History Study (RNHS, NCT00299312, NCT02738281), a longitudinal study incorporating caregiver-provided historical and clinically observed information spanning from 2006 to 2021. A total of 1826 individuals participated in the RNHS with an average of 5 visits per individual (ranging from 1 to 18 visits). Participants enrolled had a diagnosis of RTT as well as people who did not meet RTT diagnostic criteria but had pathogenic variants in *MECP2*, and individuals with RTT-related disorders including *MECP2* duplication syndrome, CDKL5 deficiency disorder, and FOXG1 syndrome. For this study, we only included those individuals with a diagnosis of RTT (Classic or Atypical RTT). All participants provided genetic testing results.

To characterize the differences in younger vs. older individuals with RTT, participants were divided into cohorts assessed under 30 years old (yo) and those assessed ≥30 yo. We excluded one individual with Classic RTT who had a mutation in *SHANK3*. For atypical RTT (*n* = 211), only 15 participants were assessed ≥30 yo, so data from individuals with Atypical RTT was not included in the final analyses. Subsequently, a total 1253 participants with Classic RTT and pathological *MECP2* loss of function variants were analyzed. Of these, 1195 had visits at ages less than 30 yo, with 1143 seen only when less than 30 yo. One hundred and ten participants were assessed when ≥30 yo, of these 58 had baseline visits ≥30 yo and an additional 52 participants seen at baseline visits under 30 yo but aged to ≥30 yo during the study (Table 1). Visits ranged from one to fourteen specific occurrences for the subset of all participants meeting these criteria and included in the analysis and results presented here. The mean age at last visit in the <30 yo cohort was 13.7 yo (SD, Range: 7.6 yo, 1.9–29.8 yo) compared to mean age at first visit in the ≥30 yo which was 34.5 yo (SD, Range: 6.1 yo, 30.0–66.5 yo). Five of the women over 30 years of age died during the RNHS, accounting for 4.5% of the 111 women with Classic RTT in this study. This was similar to the previously reported 3.9% death rate for all individuals in the RNHS.

### 2.2. Assessments

Participants were assessed in a structured in-person clinical research visit (lasting ~1–2 h), which occurred longitudinally at pre-defined intervals based on age of enrollment, ranging from yearly to every other year. In-person evaluations utilizing structured research forms including caregiver completed history and assessment forms and questionnaires, clinical histories, structured clinical exams, and clinician-completed rating scales. Clinical assessment and rating scales were conducted by physician investigators who were trained on the conduct of the study and completion of the forms via in person training at the initiation of the study or the site by the PI of the study (AKP). Clinician-rated assessments included the Clinical Severity Score (CSS) and the Motor Behavioral Assessment (MBA), two RTT specific rating sales that were used throughout the RNHS [30]. The CSS is a clinical rating scale composed of 13 items, each with a Likert Scale from 0–4 or 0–5 (higher numbers representing more severely affected), with a range of total CSS score from 0–58 (0 = unaffected, 58 = most severely affected). The MBA is a clinical rating scale composed of 34 items, each with a Likert Scale for each item from 0–4 (higher numbers representing more severely affected), and a range of total MBA score from 0–136 (0 = unaffected, 136 = most severely affected).

### 2.3. Statistical Analyses and Data Visualization

SPSS v.29.0.0.0 (IBM, Armonk, NY, USA) was used for statistical analyses and graphical representation. Data are presented as mean values with standard error of the mean (SEM), median, or percentage as appropriate, and *p*-values ≤ 0.05 are considered significant. Comparison of the frequency of specific common pathogenic *MECP2* variants between the young cohort (<30 yo) vs. the old cohort (≥30 yo) was conducted using the Fisher exact test. Difference in continuous variables (total CSS or MBA scores) between the last visit in the young cohort and the first visit in the old cohort were analyzed using one-way ANOVA (factor: age cohort). Comparison of individual items on the CSS or MBA between the last visit in the young cohort and the first visit in the old cohort were analyzed using the Kruskal–Wallis test. Evaluation in the change between the first and last visit in individuals within the young cohort or the old cohort was conducted using paired *t*-tests for continuous variables (total CSS or MBA score) or using paired Wilcoxon signed rank test for non-continuous variables (individual items in the CSS or MBA).

## 3. Results

### 3.1. Distributions of Mutations and Severity in under 30 yo vs. over 30 yo Cohorts

We hypothesized that longevity in older women would be associated with *MECP2* variants associated with overall milder involvement (R133C, R294X, R306C, and C-terminal truncations [CTT]), vs. more severe variants (R106W, R168X, R255X, R270X and large deletions [LargeDel]). However, no significant changes were noted in the proportions of variants between the two cohorts except for a significant enrichment of the relatively more severe variant R106W (3.2% < 30 yo vs. 8.2% ≥ 30 yo, *p* = 0.014, Fisher’s exact test) in the older cohort (Table 2). Therefore, our initial hypothesis that milder mutations would be overrepresented in the older cohort was not confirmed.

Furthermore, no differences were noted in total CSS or MBA scores between <30 yo cohort vs. ≥30 yo cohort for the entire group (All, Table 2). However, within specific *MECP2* variant groups, we found changes in the total CSS or MBA scores between <30 yo cohort vs. ≥30 yo cohort. In individuals with the R255X variant (a severe variant), both the total CSS and MBA scores showed increased severity in the ≥30 yo cohort. In contrast, CSS decreased in severity in individuals with the R270X variant (a severe variant), and MBA increased in severity in individuals with the R306C variant (a mild variant). Thus, we did not identify that there was an enrichment in more mildly affected individuals in the ≥30 yo cohort, nor did we find that there was any consistent finding that individuals with severe *MECP2* variants who survived over 30 yo were less affected than those with the same variants under 30 yo or the converse that individuals with mild *MECP2* variants who survived over 30 yo were less affected than those with the same variants under 30 yo.

### 3.2. Differences in Clinical Features between under 30 yo vs. ≥30 yo Cohorts

Although overall severity between the young vs. the old cohort was not different, we evaluated whether there were differences between age cohorts of specific clinical features through analyses of individual items in the CSS and MBA. The features analyzed did not include historical items (e.g., onset of regression, onset of stereotypies, head growth) or lower priority and relatively subjective items (e.g., excluded overly active/passive, toileting, self-mutilation, pain tolerance, biting, truncal rocking, myoclonus and hyperreflexia) but focused on items representing highly relevant clinical features in RTT.

Each item in the CSS and MBA is scored over a range of 0 (feature not present) to 4 or 5 (feature continuously present or severe), with the range noted for each item and the mean and median for that Item in each cohort indicated in Table 3. Significant differences were observed between last visit in <30 yo cohort and first visit in ≥30 yo cohort for several items (Table 3). The distribution of scores (Figure 1) demonstrates the differences in severity for comparisons between the younger vs. older cohorts for representative items with differences between the cohorts. Several items were more severe in the older cohort: CSS and MBA Scoliosis, CSS Nonverbal, MBA Sustained Interest, MBA Does Not Follow Verbal Commands, MBA Bradykinesia and MBA Hypertonia (Table 3 and Figure 1). Several items were less severe in the older cohort: CSS Ambulation, CSS Breathing, MBA Breath Holding, MBA Hyperventilation, MBA Mouthing Hands/Objects and MBA Stereotypies (Table 3 and Figure 1). Notably, no differences in hand skills or language were found between the last visit in <30 yo cohort and first visit in ≥30 yo cohort. So, while the overall severity scores (CSS/MBA) were not different between last visit in <30 yo cohort and first visit in ≥30 yo cohort, individual clinical features are consistent with observed clinical observations (more scoliosis, more bradykinesia/rigidity, less interactive, better breathing/hand stereotypies), but also better walking, to result in no overall differences between the two cohorts.

### 3.3. Longitudinal Progression of Overall Severity in under 30 yo vs. ≥30 yo Cohorts

The previous analyses do not necessarily indicate improvement or worsening of specific features between cohorts; rather, the cohorts are overall similar but with specific differences of clinical features. In order to evaluate whether or not longitudinal alterations of clinical severity occurred, the total CSS and total MBA from the first visit to last visit in the <30 yo cohort were compared to the first visit to last visit in the ≥30 yo cohort. The analysis was restricted to participants with more than one visit in an age cohort (*n* = 985 <30 yo cohort, *n* = 76 ≥30 yo cohort), with the average age at first visit of 8.3 yo in the <30 yo cohort vs. 34.1 yo in the ≥30 yo cohort (specifics for each group are noted in Table 4). No difference was noted in mean change in age between first-last visits in <30 yo vs. ≥30 yo cohorts (*p* = 0.517, one-way ANOVA) indicating that the individuals within each cohort are similar with regard to the longitudinal durations assessed (5.3–5.6 years). In the <30 yo cohort, both total CSS and total MBA increased from first to last visit (Table 5). Comparatively, no change was seen in total CSS and MBA in the ≥30 yo cohort from first to last visit (Table 5).

### 3.4. Longitudinal Progression of Clinical Features in under 30 yo vs. ≥30 yo Cohorts

Using a similar approach to compare the clinical features of these cohorts, specific clinical items from the CSS and MBA were compared from the first visit to last visit in the <30 yo cohort and contrasted to comparisons from first visit to last visit in the ≥30 yo cohort (Table 6). A number of clinical features worsen from first to last visit in the <30 yo group, such as in gross motor function (CSS Sitting, CSS Ambulation, MBA Motor Skills), fine motor skills (CSS Hand Use, MBA Does Not Reach for Objects/People, MBA Hand Clumsiness), verbal communication (CSS Language, MBA Verbal Skills, MBA Speech Disturbance), oro-motor function (MBA Feeding Difficulties, MBA Chewing Difficulties), musculoskeletal abnormalities (CSS Scoliosis, MBA Scoliosis, MBA Bradykinesia, MBA Dystonia, MBA Dyskinesias, MBA Hypertonia), and seizures (CSS Seizures, MBA Seizures). These findings are as expected given the clinical progression observed in people with RTT in younger ages, such as on-going reduction of hand use [31] and ambulation. In contrast, some clinical features improved in the <30 yo cohort from first visit to last visit, such as in nonverbal communication (CSS Nonverbal Communication, MBA Does Not Follow Verbal Commands), behavior (MBA Irritability or Tantrums, MBA Aggressiveness), and features such as teeth grinding, saliva expulsion, and hand mouthing (MBA Bruxism, MBA Air/Saliva Expulsion, MBA Mouthing Hands/Objects). Again, these are consistent with observed age-related changes in younger individuals with RTT.

Within the older cohort (≥30 yo), a number of clinical features were unchanged from the first to last visit (Table 6), similar to previous reports indicating that clinical features are stable in older individuals with RTT [32]. However, a number of clinical features were different between first to last visit in ≥30 yo cohort (Table 6). Worsening for the ≥30 yo was observed for gross motor skills (MBA Motor Skills, CSS Ambulation trend *p* = 0.089), fine motor abilities (CSS Hand Use, MBA Does Not Reach for Objects/People), verbal communications (CSS Language, MBA Verbal Skills, MBA Speech Disturbance), oro-motor function (MBA Chewing Difficulties), and movement disorders (MBA Hand Stereotypies, MBA Dystonia, MBA Dyskinesia). Interestingly, nonverbal communication (CSS Nonverbal Communication, MBA Does Not Follow Verbal Commands) showed improvement from the first to last visit in the ≥30 yo cohort. Representative distributions of CSS or MBA items that had significant change from first to last visit in the ≥30 yo cohort are shown in Figure 2 displays representative distributions of selected CSS items that show change from first to last visit in both the <30 yo and ≥30 yo cohorts, with CSS Ambulation, CSS Hand Use, and CSS Language showing progressive worsening in both the younger and older cohorts (Figure 2A–C), and CSS Nonverbal Communication showing progressive improvement in both cohorts (Figure 2D). Overall, while the progressive worsening in some clinical features is expected in the <30 yo group, these results demonstrate that some clinical features improve in the <30 yo group and in contrast to previous reports, older people with RTT continue to show changes in clinical features with ongoing worsening of many functional skills, but improvement in nonverbal communication.

## 4. Discussion

Prolonged survival of individuals with RTT has been known for more than fifteen years [8], but the specific features of those surviving past thirty years has received scant attention. The RNHS evaluated more than 1600 girls and women with RTT over the past 16 years and among this group we analyzed 1253 with classic RTT; 1143 were under 30 yo at the last visit (mean age 13.9 yo), while 110 were ≥30 yo at their last visit (mean age 39.4 yo). The presumption was that maintenance of ambulation and purposeful hand function and lesser difficulties with seizures and scoliosis associated with specific point mutations (R133C, R294X, R306C and CTT) would influence the likelihood of longevity preferentially.

Contrary to our expectation, we did not see a notable difference in the distribution of specific point mutations between the younger and older cohorts, arguing against increased survival specifically in individuals with mild mutations. In other words, older age people with RTT were not dominated by mild *MECP2* mutations or overall decreased severity. Additionally, no difference in overall severity was noted between these two groups. However, an ongoing progression of clinical features with worsening functional skills loss and motor features but improvement in nonverbal communication with age was observed. Overall, this indicates a further need to improve our understanding of age-related progression in RTT. Other factors must underlie the inability to support our initial hypotheses. Improved nutrition, better management of epilepsy, pulmonary, and gastrointestinal issues, consistent physical, occupational, and communication therapies, and better attention to orthopedic issues such as scoliosis and joint deformities all promote better health and could be a reason for improved longevity in all variant groups. Further, overall increase in experience of child neurologists, geneticists, and primary care physicians with RTT has advanced the care of these issues during the past twenty or more years. The improvement in care in these domains could be the explanation for prolongation of survival and are areas for further research in the future. Environmental conditions could also be important factors affecting longevity. Most individuals with RTT in the US are cared for in their own homes where enrichment is more likely. Nevertheless, once individuals with developmental issues such as RTT age out of school-based programs, typically at age 22 years, their ability to access programs that increase socialization and provide quality therapeutic programs becomes increasingly more difficult. Differences in access to these programs represent another possible reason for better survival. Indeed, animal studies have shown that clear differences in outcome are related to the quality and quantity of environmental factors including environmental enrichment and socialization [33,34,35]. Although evidence on the role of other genetic factors is lacking, these could be at play as well.

While it had been thought that clinical features are stable after 30 yo in RTT, progressive worsening of functional skills such as hand use, ambulation, and speech, and worsening of features such as chewing, feeding, dystonia, dyskinesia, and decreased sustained interest may occur. However, this is accompanied by stability in many clinical domains such as epilepsy and improvements in nonverbal skills. It is important for clinicians and caregivers to recognize the progression of these clinical features with age. Awareness of this important aspect is vital for discussions with families both at the time of diagnosis when young, but also during transition of care from pediatricians and pediatric subspecialists to their adult counterparts. Knowing that these RTT women can live long lives means our communities need to keep focused on providing high-quality medical and allied health needs and ensuring enrichment programs remain accessible throughout their lives. Importantly, as therapies become available, caregivers and providers should ensure these individuals have access to any new therapeutic modalities. Indeed, the recent approval of trofinetide (Daybue) offers a specific oral therapy for individuals with RTT [36,37]. Other agents are currently under study and two gene therapy programs (Taysha Gene Therapies-NCT05606614 and Neurogene-NCT05898620) have now begun.

Limitations to this study are noted. While representing the largest older cohort to date, the overall number of people evaluated in the older cohort is still relatively small and there remains a need for further characterization of the longitudinal progression with age in RTT. Additionally, there may be an ascertainment bias for people who have RTT and are ≥30 yo, because the disease was not recognized as a unique disorder until after the publication by Hagberg et al. in 1983 [2], and widespread awareness of the disorder only emerged after the discovery of its genetic basis in 1999 [3]. Thus, the majority of people who currently are identified with the diagnosis are under 30 years old. This issue is further compounded by the fact that the awareness of RTT is mainly concentrated in pediatric providers, with adult providers having limited knowledge of RTT and therefore less likely to make the diagnosis of RTT in older individuals. Additionally, the people with RTT ≥30 yo who were evaluated in this study may not reflect the entire spectrum of the affected population in this age range, as they may have multiple other factors that allowed participation in this study, such as higher family socio-economic status, living near a tertiary academic center with access to diagnostic and research opportunities and overall better health. Our ability to capture all individuals with RTT in future “real-world” clinical studies will permit better evaluation of these factors. Finally, a noted limitation of this study is that it only includes information obtained from people living in a high-resource country with access to advanced medical care. As RTT is caused by spontaneous, de novo genetic variants in RTT, it occurs with equal frequency across ethnic populations and geographical areas. However, differences in access to medical care and awareness of the disorder in under resourced countries limit our knowledge of the disorder in these countries and the ability to generalize the results presented here to those regions.

## Figures and Tables

**Figure 1 genes-15-01107-f001:**
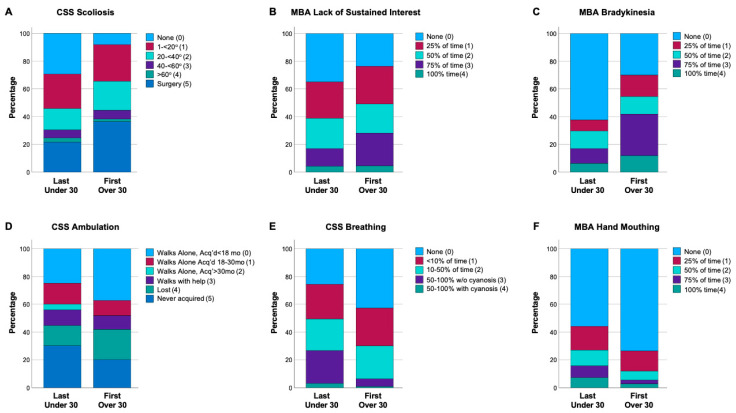
Distributions of selected CSS and MBA item scores. Differences in clinical features between last visit in <30 yo cohort are compared to first visit in ≥30 yo cohort. (**A**–**C**) display the score distribution for representative items from the CSS or MBA that were increased in severity on the older (≥30 yo) compared to the younger (<30 yo) cohort. (**D**–**F**) display the score distribution for representative items from the CSS or MBA that were decreased in severity on the older (≥30 yo) compared to the younger (<30 yo) cohort. The specific items in each panel are labeled at the top of the graph, with the legend showing the item score responses and color labels. Graphs show the percentage of each item score response, with the least severe (score 0, light blue) on top to most severe score (score 4, dark green; or score 5, dark blue) on bottom.

**Figure 2 genes-15-01107-f002:**
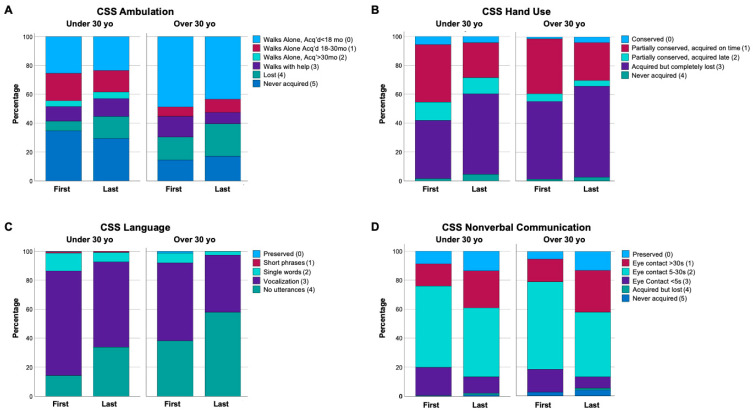
Change in individual participant’s clinical features between first to last visit in younger (<30 yo) and older (≥30 yo) cohorts. (**A**–**C**) display clinical features that show progressive worsening from first to last visit in both cohorts. (**D**) displays a clinical feature (nonverbal communication) that shows progressive worsening from first to last visit in both cohorts. The specific items in each panel are labeled at the top of the graph, with the legend showing the item score responses and color labels. Graphs show the percentage of each item score response, with the least severe (score 0, light blue) on top to most severe score (score 4, dark green; or score 5, dark blue) on bottom.

**Table 1 genes-15-01107-t001:** Number of participants with Classic RTT in each cohort.

Classic RTT	Number
All Participants	1253
Visits < 30 yo (all)	1196
Only seen < 30 yo	1143
Visits ≥ 30 yo (all)	110
Baseline visits ≥ 30 yo	56
Aged >30 yo during study	52

**Table 2 genes-15-01107-t002:** Severity Scales and distribution of MECP2 mutations between cohorts. Total severity measures using the total CSS or total MBA were compared between the last visit <30 yo cohort vs. first visit ≥30 yo cohort analyzed via one-way ANOVA (factor: cohort), with the mean and standard error of the mean (SEM) presented. Difference indicates the numerical difference between the values for the ≥30 yo cohort and the <30 yo cohort. Specific mutation frequency (percentage) between young and old cohorts analyzed using Fisher Exact Test. *p*-values are displayed, with significant values shown in bold text.

Mutation	Age Group	*n*	CSS			MBA			Mutation Percentage		
			Mean (SEM)	Change	*p*-Value	Mean (SEM)	Change	*p*-Value	Percentage	Change	*p*-Value
**All**	<30	1195	24.5 (0.2)			50.1 (0.4)			NA	NA	NA
	≥30 yo	110	24.5 (0.7)	0	0.985	52.3 (1.2)	2.2	0.100	NA	NA	NA
**R106W**	<30	38	26.5 (1.1)			53 (1.9)			3.2		
	≥30 yo	9	22 (2.3)	−4.5	0.080	50.8 (3.9)	−2.2	0.616	8.2	**5.0**	**0.014**
**R133C**	<30	74	20.1 (0.8)			46.8 (1.4)			6.2		
	≥30 yo	6	17.8 (2.8)	−2.3	0.433	42.8 (4.7)	−4	0.422	5.5	−0.7	1.000
**R168X**	<30	127	27.7 (0.6)			52.9 (1.2)			10.6		
	≥30 yo	11	29.8 (2.1)	2.1	0.335	59.5 (4.1)	6.6	0.126	10.1	−0.5	1.000
**T158M**	<30	126	24.8 (0.7)			50.5 (1.2)			10.5		
	≥30 yo	8	26.5 (2.6)	1.7	0.537	54.6 (4.6)	4.1	0.383	7.3	−3.2	0.328
**R255X**	<30	115	27.2 (0.6)			51.7 (1)			9.6		
	≥30 yo	5	35.4 (3.1)	**8.2**	**0.010**	64.2 (4.9)	**12.5**	**0.013**	4.6	−5.0	0.085
**R270X**	<30	72	27.6 (0.9)			53 (1.5)			6.0		
	≥30 yo	8	22.1 (2.6)	**−5.5**	**0.049**	49.1 (4.4)	−3.9	0.407	7.3	1.3	0.537
**R294X**	<30	71	21 (0.8)			48.8 (1.6)			5.9		
	≥30 yo	11	20.6 (1.9)	−0.4	0.878	45.5 (4.1)	−3.3	0.449	10.0	4.1	0.100
**R306C**	<30	94	20.3 (0.7)			46.3 (1.3)			7.9		
	≥30 yo	6	24.7 (2.9)	4.4	0.146	57.8 (5.3)	**11.5**	**0.038**	5.5	−2.4	0.455
**CTT**	<30	126	22.1 (0.7)			48.8 (1.2)			10.5		
	≥30 yo	13	20.7 (2.1)	−1.4	0.513	49.8 (3.8)	1	0.787	11.8	1.3	0.630
**LargeDel**	<30	108	26.3 (0.8)			52.8 (1.3)			9.0		
	≥30 yo	13	29.2 (2.2)	2.9	0.225	55.4 (3.7)	2.6	0.500	11.9	2.9	0.310

**Table 3 genes-15-01107-t003:** Comparison of clinical features between cohorts. Statistical comparisons (last visit <30 yo cohort vs. first visit ≥30 yo cohort) via Kruskal–Wallis with *p*-values are noted. Significant differences are noted in bold text.

Scale	Item	Mean: <30, ≥30	Median: <30, ≥30	Test Statistic	*p*-Value
**CSS**	Somatic Growth	1, 1.3	0, 1	2.947	0.086
	Sitting	1.3, 1.5	0, 0	0.079	0.779
	**Ambulation**	**2.7, 2.3**	**3, 3**	**4.908**	**0.027**
	Hand Use	2.3, 2.2	3, 3	0.694	0.405
	**Scoliosis**	**1.9, 2.8**	**1, 2**	**23.601**	**0.000**
	Language	3.2, 3.3	3, 3	1.795	0.180
	**Nonverbal**	**1.6, 1.9**	**2, 2**	**6.954**	**0.008**
	**Breathing**	**1.5, 0.9**	**1, 1**	**24.919**	**0.000**
	Autonomic	1, 1	1, 1	0.001	0.976
	Seizures	1.2, 1	0, 0	0.339	0.560
**MBA**	Motor Skills	2.9, 2.8	3, 3	0.681	0.409
	Verbal Skills	1.9, 2	2, 2	0.035	0.852
	Social Eye Contact	1.4, 1.5	1, 2	2.283	0.131
	**Lack of Sustained Interest**	**1.2, 1.6**	**1, 1**	**8.136**	**0.004**
	Irritability/Tantrums	0.2, 0.1	0, 0	3.622	0.057
	Does Not Reach for Objects/People	2.6, 2.6	3, 3	0	0.987
	**Does Not Follow Verbal Commands**	**1.5, 1.7**	**1, 2**	**4.428**	**0.035**
	Feeding Difficulties	1.7, 1.7	2, 1	0.09	0.764
	Chewing Difficulties	2, 1.8	2, 2	1.783	0.182
	Aggressiveness	0.1, 0.1	0, 0	0.003	0.959
	Seizures	1.4, 1.3	1, 1	0.542	0.461
	Speech Disturbance	3.2, 3.3	3, 3	1.323	0.250
	Bruxism	0.8, 0.7	0, 0	1.897	0.168
	**Breath Holding**	**1.1, 0.9**	**1, 0.5**	**10.666**	**0.001**
	**Hyperventilation**	**0.8, 0.3**	**0, 0**	**30.178**	**0.000**
	Air/Saliva Expulsion	1.3, 1.3	1, 1	0.106	0.744
	**Mouthing Hands/Objects**	**0.9, 0.5**	**0, 0**	**15.051**	**0.000**
	Hand Clumsiness	3.1, 3.1	4, 4	0.023	0.880
	**Hand Stereotypies**	**3.3, 2.9**	**4, 3.5**	**6.113**	**0.013**
	**Bradykinesia**	**0.9, 1.8**	**0, 2**	**46.333**	**0.000**
	Dystonia	1.3, 1.4	1, 1	0.391	0.532
	**Scoliosis**	**1.7, 2.4**	**1, 2**	**22.913**	**0.000**
	Dyskinesias	0.4, 0.3	0, 0	3.006	0.083
	**Hypertonia**	**1.5, 2.2**	**1, 3**	**17.349**	**0.000**
	Vasomotor Disturbance	1.1, 1.2	1, 1	2.269	0.132

**Table 4 genes-15-01107-t004:** Age of cohorts assessed longitudinally. Mean, standard deviation (SD), and range displayed for younger (<30 yo) and older (≥30 yo) cohorts for the first and last visit, and the change in age from first to last visit for both cohorts presented.

	Age Mean (SD, Range)	
	<30 yo	≥30 yo
First Visit	8.3 (6.5, 1.1–29)	34.1 (5.3, 30–57.1)
Last Visit	13.9 (7.3, 2.4–30)	39.4 (7, 31.1–64.5)
Age Change (First-Last)	5.6 (3.7, 0.4–15.3)	5.3 (3.4, 1–13.7)

**Table 5 genes-15-01107-t005:** Change in total CSS and MBA between first and last visits in <30 yo and ≥30 yo groups. Changes in total CSS and MBA were analyzed using paired *t*-tests with *p*-values noted. Significant differences are noted in bold text.

Age Group	Number	Visit	CSS			MBA		
			Mean (SEM)	Change	*p*-Value	Mean (SEM)	Change	*p*-Value
<30	985	First	22.2 (0.2)			46.9 (0.4)		
		Last	24.6 (0.2)	2.5	**0.000**	50.7 (0.4)	3.8	**0.000**
≥30 yo	76	First	23.3 (0.9)			51 (1.5)		
		Last	23.5 (0.8)	0.2	0.573	52.6 (1.4)	1.6	0.151

**Table 6 genes-15-01107-t006:** Change in clinical features between first-to last visit in <30 yo and ≥30 yo groups. Differences in clinical features from first to last visit were analyzed using Wilcoxon signed rank (paired) with the test statistic and *p*-values noted. Significant differences are noted in bold text.

	Item	Age Group	Mean: First, Last	Median: First, Last	Test Statistic	*p*-Value
**CSS**	**Somatic Growth**	<30	1, 0.9	0, 0	−1.608	0.108
		≥30	1.3, 1.1	0, 0.5	−1.869	0.062
	**Sitting**	<30	**1, 1.3**	**0, 0**	**6.265**	**0.000**
		≥30	1.1, 1.3	0, 0	1.252	0.210
	**Ambulation**	<30	**2.6, 2.7**	**3, 3**	**3.208**	**0.001**
		≥30	1.9, 2.1	1, 1	1.701	0.089
	**Hand Use**	<30	**1.9, 2.3**	**2, 3**	**12.461**	**0.000**
		≥30	**2.2, 2.3**	**3, 3**	**2.265**	**0.024**
	**Scoliosis**	<30	**1, 2**	**0, 1**	**18.891**	**0.000**
		≥30	2.6, 2.7	2, 2	0.959	0.337
	**Language**	<30	**3, 3.3**	**3, 3**	**10.921**	**0.000**
		≥30	**3.3, 3.6**	**3, 4**	**3.162**	**0.002**
	**Nonverbal Communication**	<30	**1.9, 1.6**	**2, 2**	**−7.140**	**0.000**
		≥30	**2, 1.7**	**2, 2**	**−2.211**	**0.027**
	**Respiratory Dysfunction**	<30	**1.3, 1.5**	**1, 2**	**5.114**	**0.000**
		≥30	1.1, 1	1, 1	−0.887	0.375
	**Autonomic Symptoms**	<30	**0.9, 1**	**1, 1**	**4.388**	**0.000**
		≥30	0.9, 0.9	1, 1	−0.604	0.546
	**Seizures**	<30	**0.8, 1.2**	**0, 0**	**6.132**	**0.000**
		≥30	0.9, 0.8	0, 0	−1.249	0.212
**MBA**	**Motor Skills**	<30	**2.6, 3**	**3, 3**	**12.940**	**0.000**
		≥30	**2.7, 3**	**3, 3**	**2.966**	**0.003**
	**Verbal Skills**	<30	**1.9, 2**	**1, 2**	**2.359**	**0.018**
		≥30	**2, 2.6**	**2, 3**	**3.344**	**0.001**
	**Social Eye Contact**	<30	**1.1, 1.4**	**1, 1**	**5.124**	**0.000**
		≥30	1.5, 1.6	2, 2	0.215	0.830
	**Lack Of Sustained Interest**	<30	1.2, 1.2	1, 1	−0.819	0.413
		≥30	1.7, 1.4	2, 1	−1.813	0.070
	**Irritability or Tantrums**	<30	**0.4, 0.2**	**0, 0**	**−7.609**	**0.000**
		≥30	0.2, 0.1	0, 0	−0.922	0.356
	**Does Not Reach for Objects/** **People**	<30	**1.9, 2.6**	**2, 3**	**13.481**	**0.000**
		≥30	**2.5, 3**	**3, 4**	**2.967**	**0.003**
	**Does Not Follow Verbal** **Commands**	<30	**1.8, 1.4**	**2, 1**	**−8.256**	**0.000**
		≥30	**1.8, 1.5**	**2, 1**	**−2.277**	**0.023**
	**Feeding Difficulties**	<30	**1.4, 1.8**	**1, 2**	**9.066**	**0.000**
		≥30	1.4, 1.7	1, 2	1.873	0.061
	**Chewing Difficulties**	<30	**1.6, 2.1**	**1, 2**	**11.301**	**0.000**
		≥30	**1.6, 2**	**1, 2**	**2.886**	**0.004**
	**Aggressiveness**	<30	**0.2, 0.1**	**0, 0**	**−3.243**	**0.001**
		≥30	0.1, 0.1	0, 0	−0.832	0.405
	**Seizures**	<30	**1, 1.4**	**0, 1**	**7.546**	**0.000**
		≥30	1.2, 1.1	1, 1	−1.062	0.288
	**Speech Disturbance**	<30	**2.9, 3.2**	**3, 3**	**13.835**	**0.000**
		≥30	**3.3, 3.5**	**3, 4**	**3.086**	**0.002**
	**Bruxism**	<30	**1.1, 0.7**	**1, 0**	**−7.951**	**0.000**
		≥30	0.6, 0.5	0, 0	−1.414	0.157
	**Breath Holding**	<30	1.2, 1.1	1, 1	−0.650	0.516
		≥30	1, 0.8	1, 0	−1.719	0.086
	**Hyperventilation**	<30	**0.9, 0.7**	**0, 0**	**−2.635**	**0.008**
		≥30	0.3, 0.4	0, 0	1.296	0.195
	**Air/Saliva Expulsion**	<30	**1.4, 1.3**	**1, 1**	**−2.343**	**0.019**
		≥30	1.2, 1.1	1, 1	−0.566	0.571
	**Mouthing Hands/Objects**	<30	**1.4, 0.9**	**1, 0**	**−9.066**	**0.000**
		≥30	0.4, 0.4	0, 0	−0.083	0.934
	**Hand Clumsiness**	<30	**2.8, 3.1**	**3, 4**	**9.574**	**0.000**
		≥30	3.1, 3.3	4, 4	1.947	0.052
	**Hand Stereotypies**	<30	3.3, 3.3	4, 4	−1.756	0.079
		≥30	**3, 3.3**	**4, 4**	**2.012**	**0.044**
	**Bradykinesia**	<30	**0.5, 0.9**	**0, 0**	**11.131**	**0.000**
		≥30	1.6, 1.6	2, 1	−0.545	0.586
	**Dystonia**	<30	**0.8, 1.4**	**0, 1**	**12.119**	**0.000**
		≥30	**1.4, 1.8**	**1, 2**	**2.260**	**0.024**
	**Scoliosis**	<30	**0.9, 1.7**	**0, 1**	**18.300**	**0.000**
		≥30	2.2, 2.3	2, 2	0.786	0.432
	**Dyskinesias**	<30	**0.3, 0.4**	**0, 0**	**2.679**	**0.007**
		≥30	**0.2, 0.5**	**0, 0**	**2.128**	**0.033**
	**Hypertonia**	<30	**0.9, 1.6**	**0, 1**	**12.858**	**0.000**
		≥30	2.1, 2.4	3, 3	1.637	0.102
	**Vasomotor Disturbance**	<30	**1, 1.1**	**1, 1**	**2.318**	**0.020**
		≥30	1.1, 1	1, 1	−1.139	0.255

## Data Availability

The datasets from the Rett syndrome and Rett-related Disorders Natural History Study (RNHS) have been deposited to the database of Genotypes and Phenotypes (dbGAP) repository, phs000574.v1.p1 and hyperlink to dataset(s) in https://www.ncbi.nlm.nih.gov/projects/gap/cgi-bin/study.cgi?study_id=phs000574.v1.p1 (accessed on 1 January 2023), and deposited to dbGAP per a predefined schedule at regular intervals. Additionally, datasets used for the analysis conducted within this work are available from the corresponding author on reasonable request and pursuant to any required data transfer and use agreements.

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
