# Peer review of "Clinical Features and Disease Progression in Older Individuals with Rett Syndrome"

_genes, 2024, doi:10.3390/genes15081107_

Round 1

Reviewer 1 Report

Comments and Suggestions for Authors

In this study by Neul et al. the authors compared the pathological and phenotypic features of individuals with Rett syndrome and harboring various forms of mild to severe mutant variants of MECP2 protein between <30 and >30 years old subjects. While this study is important for the genetic counseling of Rett patients, however, the study lacks some critical merits for being considered for publication in its present form in this journal. The authors may consider the following points to improve the manuscript further for resubmission to this journal

1) Line 82 states "ranging from 1 to 18 visits" whereas Line 97 states "Visits ranged from one to fourteen specific occurrences". Please correct this inconsistency.

2) Did the authors perform in-person evaluations for 1-2 h for participants aged 1.9 - 10 years in an exactly similar fashion to adults? For the <30 yo group,  I am afraid that the inclusion of children and teenagers is likely to influence the cohort outcome. Can the authors shed light on this point? If there was any additional variability to be considered in this study.

3) Also among the children, did the Rett syndromes were stabilized or still developing to manifest at the time of study?

4) Line 146 - "withing" should be "within".

5) Between lines 146 - 151, it is not clear why the severe variants like R255X and R270X showed opposite severity levels. 

6)Between lines 146 - 157, it is not understood why different variants are behaving differently in <30 or >30 years old individuals, suggesting that there could be other factors that decide the course of the disease severity. In the same sense, it indicates that the current study is in its very preliminary stage to be published.

7) Another major drawback of this study could be disproportionate group sizes of >30 and <30 yo cohorts for R106W, R255X and R270X sub-groups.

8) Between lines 336 - 341, the meaning is not clear.

9) In the discussion section, can the authors link any majorly observed MECP2 mutation to motor function-related pathway defects?

10) Since the average age of the >30 yo group was 34.5 years, it is not logical to mention "older individuals" in the title. The authors may consider replacing it with middle-aged individuals unless they have a plan to include additional >50 yo age data to this study.

11) Please include all the major drawbacks of this study in the limitation section.

Comments on the Quality of English Language

Sentence constructions are often very complicated and confusing. The whole manuscript needs to be thoroughly checked by a native speaker.

Author Response

Comment 1) Line 82 states "ranging from 1 to 18 visits" whereas Line 97 states "Visits ranged from one to fourteen specific occurrences". Please correct this inconsistency.

Reply: The discrepancy noted by the reviewer reflects the differences in the range of visits for all participants (in line 82) to the range of the number of visits for the subset of participants whose information was used for this study (line 97). We added a clarifying line in line 98 to reflect this.

Comment 2) Did the authors perform in-person evaluations for 1-2 h for participants aged 1.9 - 10 years in an exactly similar fashion to adults? For the <30 yo group, I am afraid that the inclusion of children and teenagers is likely to influence the cohort outcome. Can the authors shed light on this point? If there was any additional variability to be considered in this study.

Reply: In this study, all evaluations performed were consistent across all age ranges. While there are likely some differences in the disease features and progression in younger participants (under 10 years old) compared to people between 10-20 years old, the focus of this manuscript was the characterization of the mutation frequency between older individuals (>30 years old) to those people <30 years old. Characterization of these differences and evaluation of other factors is a valuable line of future research, but beyond the scope of this manuscript.

Comment 3) Also among the children, did the Rett syndromes were stabilized or still developing to manifest at the time of study?

Reply: The understanding of disease progression in RTT provides information on this issue, and has been characterized in greater detail in previous published manuscripts. Broadly, the dramatic loss of skills occurs between 18-30 months of life, and the vast majority of people included in this study are over the age of this rapid regression. However, the onset of some co-morbidities, such as seizures, occurs later in life (~4-5 years old), and a key point of this manuscript is that disease progression and evolution is not static in older individuals (>30 yo)

4) Line 146 - "withing" should be "within".

Reply: Thank you for identifying this, we have made the correction in the revised manuscript (line 48 in revised manuscript).

5) Between lines 146 - 151, it is not clear why the severe variants like R255X and R270X showed opposite severity levels.

Reply: We agree that it is not clear why individuals with different severe MECP2 variants display different progression in severity in the older cohort, but individual-level variation is the likely contributory factor.

Comment 6)Between lines 146 - 157, it is not understood why different variants are behaving differently in <30 or >30 years old individuals, suggesting that there could be other factors that decide the course of the disease severity. In the same sense, it indicates that the current study is in its very preliminary stage to be published.

Reply: The main point is that there is no evidence supporting the hypothesis that individuals with milder MECP2 variants, or overall less clinical severity, were more likely to survival compared to more severely affected individuals. The contribution of other factors involved in the course of disease progression and severity is an important avenue of research, that is being evaluated, but beyond the aims and scope of this current study. We disagree with the reviewer’s ascertainment that this is in preliminary stages for publication. It is important to note that the data used for this study was obtained over a 15 year period and captured information from over 1200 people with Classic RTT, representing the largest and most complete data set on this rare disorder in existence. Although having a larger sample size, especially for older individuals, would provide additional and more comprehensive information, it is important to acknowledge the challenges and the very long time required to acquire such data in a rare disorder.  

7) Another major drawback of this study could be disproportionate group sizes of >30 and <30 yo cohorts for R106W, R255X and R270X sub-groups.

Reply: Please see response to previous question. In a rare disorder, the ability to have proportionate data across all age cohorts is extremely challenging in general, and it is important to note that RTT only gained clear recognition as a specific disorder in 1983, and true widespread knowledge of the disorder did not occur until after the discovery of the genetic basis of the disorder in 1999. Because of these features, there is a clear ascertainment bias for RTT, with the majority of people identified with RTT having been diagnosed after 1999, and thus are under 30 years old at this time. We provided additional explanation on this concept in the revised manuscript (lines 339-346).

Comment 8) Between lines 336 - 341, the meaning is not clear.

Reply: We have clarified this section, including expanding the limitations of the study to mention the lack of ability to generalize the findings to under-resourced countries (lines 339-358).

Comment 9) In the discussion section, can the authors link any majorly observed MECP2 mutation to motor function-related pathway defects?

Reply: Despite decades-long evaluations of transcriptional changes in neurons resulting from loss of MECP2 function, there has not emerged a clear pattern that explains motor dysfunction in RTT. Furthermore, while subtle differences have been described in transcriptional changes that arise from different MECP2 mutations, the majority of changes characterized are consistent across mutations and none have been observed that might explain differences in motor function.

Comment 10) Since the average age of the >30 yo group was 34.5 years, it is not logical to mention "older individuals" in the title. The authors may consider replacing it with middle-aged individuals unless they have a plan to include additional >50 yo age data to this study.

Reply: The median lifespan of people with RTT is roughly 50 years old. Thus, while for the general population, 30 years old would be considered “young adulthood” or “early middle age” it is considered older for people with RTT.

Comment 11) Please include all the major drawbacks of this study in the limitation section.

Reply: We expanded the final paragraph to more clearly articulate the limitations (lines 339-358).

Reviewer 2 Report

Comments and Suggestions for Authors

Article review ID3128799 GENES MDPI

In this 15-years longitudinal study of follow-up of adults with Rett syndrome, authors report the clinical evolution and aim to correlate with milder and severe genetic variants in MECP2 gene. They included 1253 participants with classic RTT and pathological MECP2 loss of function variants; however, this hypothesis was not confirmed.

SOME COMMENTS TO THE AUTHORS:

The paper is well written, and covers a relevant topic for clinicians treating this disorder and its comorbidities. However, it should be mentioned that this occurs in first-world countries where there is close clinical follow-up and a high-quality medical healthcare, and this may differ in developing countries, also considering the differences in frequencies of MECP2 gene variants depending on ethnic origin of the population studied.

- I would like the authors to comment briefly on this.

- They observed some differences on clinical scales for specific variants, but nothing conclusive. Then they analyzed specific items of CSS/MBA, clinically relevant for RTT. They found that several items were more severe in the older cohort, but at the same time they observed several items were less severe in this same cohort. Therefore, in my opinion, there must be too many other factors modulating evolution of RTT that not were considered in this study. Please, comment on this (genetic or epigenetic modifiers, environmental and social factors).

-Were there additional comorbidities not presented in the paper in both cohorts (e.g., pneumonia, heart failure, gastric ulcer, and others)?

-What are main contributions of your paper?

-What should clinicians caring for patients with RTT pay more attention to?

Author Response

Comment 1: The paper is well written and covers a relevant topic for clinicians treating this disorder and its comorbidities. However, it should be mentioned that this occurs in first-world countries where there is close clinical follow-up and a high-quality medical healthcare, and this may differ in developing countries, also considering the differences in frequencies of MECP2 gene variants depending on ethnic origin of the population studied. I would like the authors to comment briefly on this.

Reply: The reviewer raising an important point related to the fact that our study presents data derived from individuals in first-world countries, who likely have different disease progression and survival based on increased access to medical care and other support systems. Because the Rett syndrome is caused by de novo mutations in MECP2, which occur at the same rate across ethnic groups and geographical distributions, the expectation is that there should be a consistent incidence of Rett syndrome across the world. This has been demonstrated to be true in countries where it has been studied, such as in China and India, however there is a lack of systematic information from other areas of the world such as in sub-Saharan Africa. The reasons for this are not likely due to variation in the incidence, but likely reflect decreased awareness and an overall lack of access to health care in those areas. Unfortunately, because of these features, the ability to describe and comment on the differences in disease progression and survival for people with Rett syndrome in first-world countries compared to countries lacking such resources is not possible at this time. We hope that increased awareness of Rett syndrome and access to care will provide data to enable these types of comparisons. We have added a sentence within the limitations section addressing these issues (lines 351-357).

Comment 2: They observed some differences on clinical scales for specific variants, but nothing conclusive. Then they analyzed specific items of CSS/MBA, clinically relevant for RTT. They found that several items were more severe in the older cohort, but at the same time they observed several items were less severe in this same cohort. Therefore, in my opinion, there must be too many other factors modulating evolution of RTT that not were considered in this study. Please, comment on this (genetic or epigenetic modifiers, environmental and social factors).

Reply: There are likely a number of additional factors, ranging from genetic and genetic to social and environmental, that influence both the disease progression and overall clinical condition of people with Rett syndrome. Identifying and understanding these issues is a critical area of research, one challenged by the rare nature of the disorder and beyond the scope of the work presented in this manuscript.

Comment 3: Were there additional comorbidities not presented in the paper in both cohorts (e.g., pneumonia, heart failure, gastric ulcer, and others)?

Reply: The contribution of other common comorbidities found in people with Rett syndrome, such as sleep disturbances, gastroesophageal reflux, and constipation were not directly evaluated within this manuscript; however, the rating scales used in this study include high-frequency co-morbidities found in Rett syndrome, notably seizures, breathing irregularities, chewing and swallowing problems, behavioral problems, vasomotor problems, etc (see Table 3). Thus, a significant number of common co-morbidities found in Rett syndrome were analyzed, and future evaluations can address other clinical issues not characterized in this manuscript.

Comment 4: What are main contributions of your paper?

Reply: The first main contribution is the refutation of the widely held belief that increased survival is more likely in people with RTT who have milder mutations compared to people who have severe mutations. The second main contribution is the importance of the recognition that, despite prevailing anecdotal opinions, the clinical features found in people with RTT are not static in older age individuals. Thus understanding is important for clinicians caring for people with RTT, as they should be aware of the ongoing evolution of the disorder and specifically consider and routine evaluation those features that worsen with age in order to develop clinical care plans addressing these problems.

Comment 5:What should clinicians caring for patients with RTT pay more attention to?

Reply: Please see previous answer.
